# Investigation of new inflammatory biomarkers in patients with brucella

**Revşa Evin Canpolat Erkan** [1]*, **Recep Tekin** [2]

1 Department of Clinical Biochemistry, Faculty of Medicine, Dicle University, Diyarbakir, Turkey,
2 Department of Infectious Diseases and Clinical Microbiology, Faculty of Medicine, Dicle University, Diyarbakir, Turkey

* drevinerkan@gmail.com

## Abstract

### Background

Delayed diagnosis and inadequate treatment of infectious and inflammatory diseases, such as Brucella, lead to high rates of mortality and morbidity. The aim of our study was to investigate the association between serum levels of apelin, presepsin, and irisin with inflammation, laboratory parameters, and blood culture in patients with brucella.

### Patients and methods

This prospective case-control study involves 30 patients with brucellosis and 30 healthy, matched control subjects. Thirty patients who were diagnosed with brucellosis were aged ≥ 18 years. Blood samples were taken from the patients on the first day they were diagnosed with brucellosis. The values of irisin, presepsin, and apelin were studied. In addition, blood samples were also taken from 30 healthy individuals for the control group. Irisin, presepsin, and apelin values that were measured in the patients on the first day were compared with those values measured in the control group.

### Results

The sex and age statuses of the subjects are matched among the groups. The levels of irisin were significantly higher in patients with brucellosis compared to the control group (p<0.045). There was no significant difference between the two groups in terms of apelin and presepsin levels (p values 0.087 and 0.162, respectively). There was a positive correlation between irisin levels and elevated ALT levels, as well as positive blood cultures.

### Conclusions

It appears that the measurement of irisin levels may be beneficial in patients with brucellosis. Irisin can be used as a diagnostic marker for brucella infection and may greatly clinicians to predict the severity disease and treatment response.

**Data Availability Statement:** Biochemical and microbiological examination data of the patients included in the study were obtained using the data of the university hospital. Since the university hospital is a public institution, patient data is not publicly available. The entire underlying dataset can

be obtained from the Ethics Committee. Ethics committee e-mail: kuruletikdiyar@gmail.com The e-mail address of Ethics Committee Chairman İbrahim Kaplan is dribrahimkaplan@hotmail.com. Telephone number of the institution: +90 412 248 80 01/ 5284.

**Funding:** This study was supported by the Dicle University Scientific Research Projects Commission with the project number TIP.20.037. The funders had no role in study design, data collection and analysis, decision to publish, or preparation of the manuscript.

**Competing interests:** The authors declare there are no conflicts of interests.

## Introduction

Brucellosis is a zoonotic disease caused by bacteria of the genus Brucella, which is typically transmitted from infected animals [1–3]. Delayed diagnosis and inadequate treatment of infectious and inflammatory diseases, such as Brucella, result in high mortality and morbidity rates, as well as significant increases in national health expenditures. Therefore, there has been a continuous search for diagnosing these patients and assessing treatment responses [4–6]. C-reactive protein, sedimentation, and complete blood count are common tests routinely requested by physicians to diagnose infectious and inflammatory diseases, determine their severity, and evaluate the response to treatment. However, in endemic areas, these laboratory findings are sometimes insufficient to diagnose brucellosis. Additionally, the persistence of high brucella antibody titers at the end of treatment or even for months after treatment can pose challenges for physicians in determining when to terminate treatment or diagnose relapse [7–9]. Molecular biomarkers, such as irisin, presepsin, and apelin, which have been extensively studied in recent years, have been found to significant play a role in the development of various diseases. Irisin, presepsin, and apelin are also believed to play a role in inflammatory responses [10–12]. Various molecules and enzymes are studied for this purpose. In recent years, new methods have become increasingly important in determining the progression of brucellosis. The aim of this study is to determine inflammatory parameters in patients with brucella by using non-invasive methods. By obtaining information about the molecules involved in inflammation in brucellosis patients, it is possible to make predictions regarding the diagnosis, treatment, and prognosis of these patients. Overall, the analysis of specific markers in the serum sample can help in assessing the severity and impact of challenging-to-diagnose and chronic brucella infections. These markers can provide valuable information for clinicians to guide treatment decisions and monitor the progress of the infection.

## Materials and methods

This study is a single-center, prospective cohort study. In the study, we included 30 brucellosis patients who were aged $\geq$ 18 years and were hospitalized in our center for a period of two years. Those patients diagnosed with Brucella (culture positive and/or Rose bengal positive and clinically compatible patients) between January 2021 and December 2021and healthy group were included in the study. Patients with respiratory, endocrine, cardiac, metabolic, renal or hepatic disease and pregnant women were excluded. Informed Voluntary Consent Form about the study was obtained from those included in the study. For the study, the approval of Ethics Committee was obtained from the Health Sciences University, Diyarbakır Gazi Yaşargil Training and Research Hospital. The study was conducted in accordance with the Declaration of Helsinki.

Brucellosis was diagnosed based on one of the following criteria: isolation of Brucella spp in blood, bone marrow, or cerebrospinal fluid (CSF) and other body fluids or tissue samples; a compatible clinical picture, such as arthralgia, fever, sweating, chills, headache, and malaise, supported by the detection of specific antibodies at significant titers and/or the demonstration of at least a four-fold rise in antibody titer in serum specimens taken over 2 or 3 weeks. Significant titers were defined as those equal to or greater than 1/160 in the standard tube agglutination test (STA). Serologic tests were conducted by using previously described techniques. Screening was performed by using the Rose Bengal plate agglutination test and the Brucella capture test. Simultaneously, blood samples were taken from the patients, centrifuged, and the serum was separated. It was then stored at -80 degrees Celsius in order to safely preserve the bulk quantities of irisin, presepsin, and apelin until they could be studied. In addition, blood samples were taken from 30 individuals with a body mass index (BMI) of less than 25, who

were free from any chronic diseases, did not take regular medication, and served as a control group. These samples were used to determine the values of the biomarkers apelin, presepsin, and irisin in the healthy individuals. Serum levels of Human Irisin, Human Presepsin, and Human Apelin were measured by using the BioTek ELx50 Microplate Washer and BioTek ELx800 Microplate Reader (BioTek Instruments, Inc., USA) in accordance with the instructions provided by the commercial kits (Sunred Biological Technology, Shanghai, China).

### Measurement of serum apelin

Serum prolidase enzyme activity was determined by using a commercially available quantitative enzyme-linked immune sorbent assay (ELISA) technique (Sunred Biotechnology Company, Shanghai, China) according to the manufacturer's instructions. Serum Apelin levels were shown as ng/mL.

### Measurement of serum presepsin

Serum visfatin levels were measured by using a commercial quantitative enzyme-linked immune sorbent assay (ELISA) technique (Sunred Biotechnology Company, Shanghai, China) according to the manufacturer's instructions. Serum presepsin levels as expressed as pg/mL.

### Measurement of serum irisin

Serum chemerin levels were measured by using a commercial quantitative enzyme-linked immune sorbent assay (ELISA) technique (Sunred Biotechnology Company, Shanghai, China) according to the manufacturer's instructions. Serum irisin levels were shown as ng/mL.

### Statistical analysis

The SPSS 22.0 (SPSS, IBM Corp., Armonk, NY, USA) for Windows 20.0 package program was used for statistical analysis. The Kolmogorov-Smirnov test was used to confirm that the data were within the ranges of normal distribution in both groups. A nonparametric test was employed for the variables outside the normal distribution. For the comparison of continuous parameters between reciprocal groups, the Mann-Whitney U Test or student's t test was used as an appropriate test. The chi square test was used for the analysis of categorical parameters. The Spearman or Pearson test was used for the analysis of correlation, as appropriate. Statistical significance was based on a value of $p<0.05$ with a 95% confidence interval.

## Results

The sex and age status of the subjects are matched between the groups. Among the 30 patients with brucellosis, 17 (57%) were male and 13 (43%) were female, with a mean age of 34.7 ± 16.6 years. Irisin, presepsin, and apelin values found in the blood samples taken from the patient group were compared with those values in the blood taken from the healthy control group. Overall, the levels of irisin were significantly higher in patients with brucellosis compared to the control group ($p<0.045$) (Fig 1). There was no significant difference between the two groups in terms of apelin and presepsin values (p-values of 0.087 and 0.162, respectively) (Table 1). There was a positive correlation between irisin levels and elevated ALT levels. Serum irisin concentration was significantly higher among culture-positive cases with respect to the culture-negative cases (Fig 2).

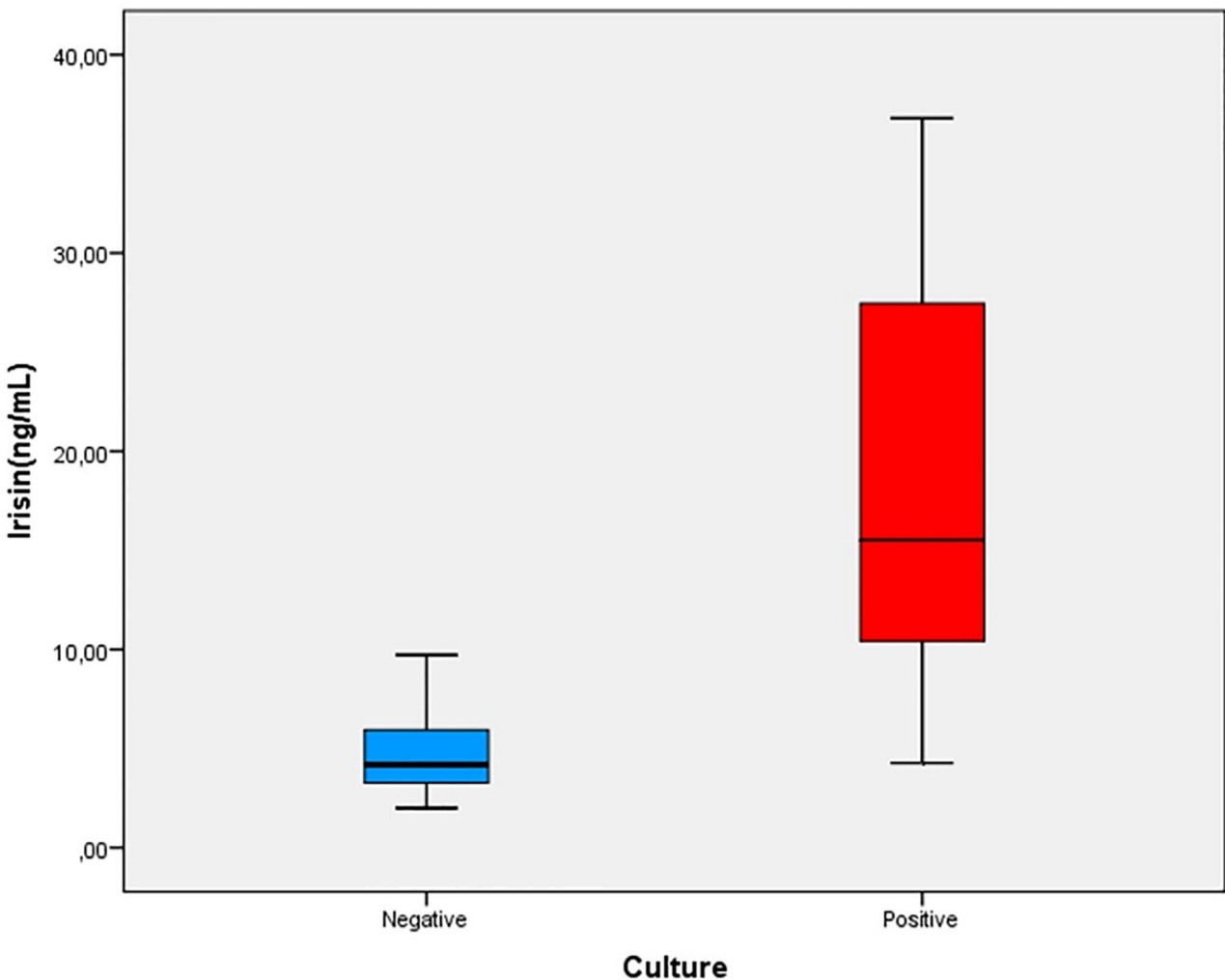

**Fig 1. Indicates overall irisin level in culture positive patients compared to culture negative patients.** Analyzed by Mann–Whitney *U* test (r = 0.719 p = 0.000).

## Discussion

The aim of this study was to quantify new inflammation biomarkers concentrations in patients with brucellosis and identify potential correlations between these biomarkers levels, blood culture, and inflammation. The initial hypothesis was confirmed, and the results revealed a positive correlation between irisin levels and blood culture, as well as a positive correlation between irisin levels and ALT levels in brucellosis patients. Blood cultures are considered the gold standard for diagnosing brucellosis [13–17]. However, in endemic areas, these laboratory

**Table 1. Irisin, presepsin and apelin level in the study groups (mean ±standard deviation).**

|  | Brucellosis patients (n = 30) | Controls (n = 30) | p |
|---|---|---|---|
| Irisin (ng/mL) | 9.27±8.8 | 4.79±5.52 | <**0.045** |
| Presepsin (pg/mL) | 1454.73 ±1231.4 | 991.61±994.78 | 0.129 |
| Apelin (ng/mL) | 22.19 ±18.47 | 20.39±17.22 | 0.494 |

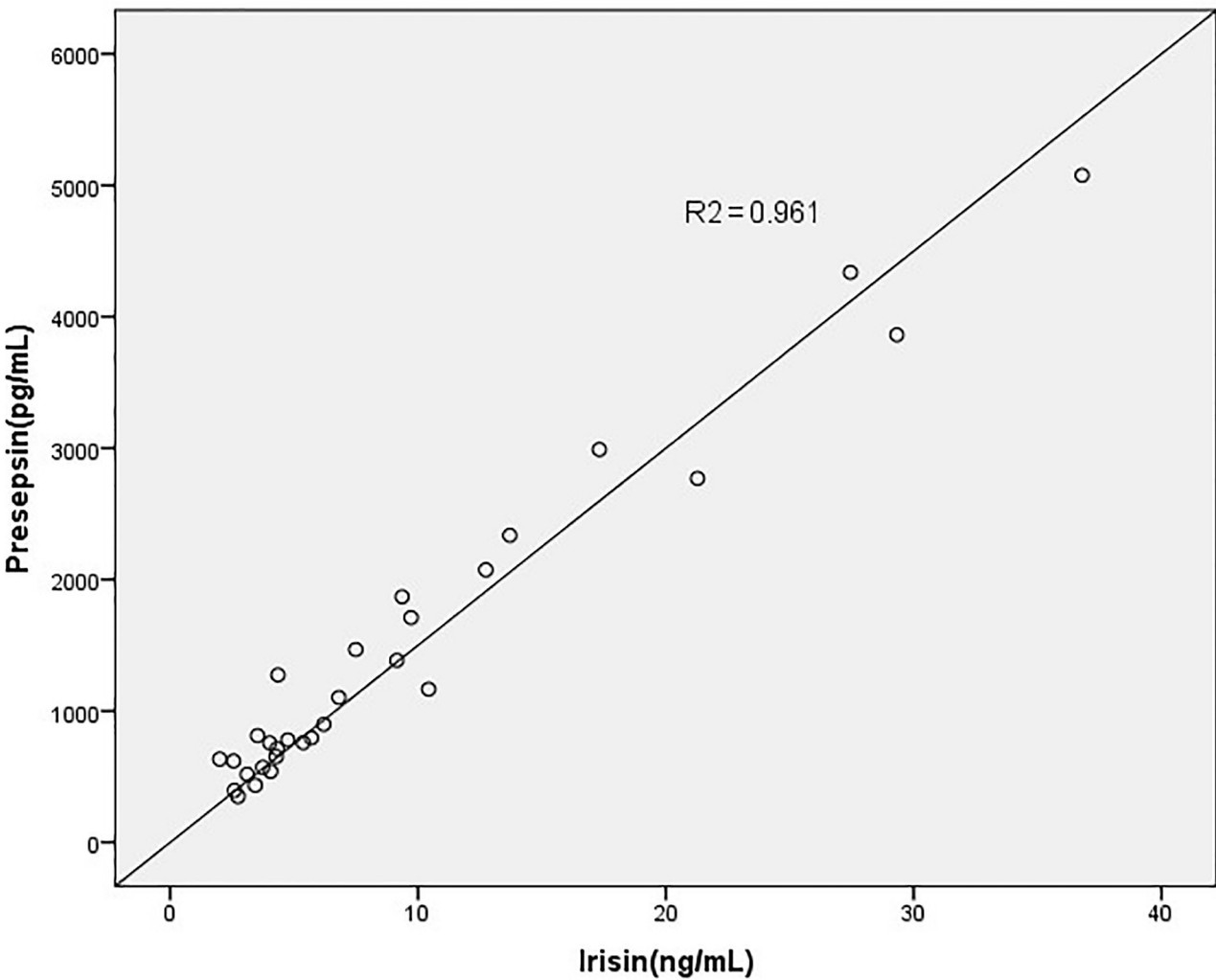

**Fig 2. Pearson correlation test indicated correlations between serum irisin level and the level of serum presepsin.**

findings are sometimes insufficient to diagnose brucellosis. Biomarkers are molecules found in blood, other body fluids, and tissues that aid in distinguishing inflammatory conditions and are utilized in the diagnosis and assessment of treatment response. Irisin, presepsin, and apelin have been studied in recent years and have been found to be involved in the pathogenesis of numerous molecular diseases.

Irisin has been reported to have important roles in energy metabolism, muscle development, neurogenesis and inflammation. Although there are many studies examining the metabolic effects of irisin, there is little data evaluating its role in immunity. The majority of those studies report that irisin has anti-inflammatory effects [10, 18]. Rizk et al. investigated serum irisin levels in adults with metabolic syndrome, and they reported that irisin levels were high in patients with metabolic syndrome and elevated liver enzymes [19]. However, there are almost no studies investigating the relationship between irisin and infectious diseases. Furthermore, Büyüktuna et al. determined a significant increase in serum irisin levels in Crimean Congo haemorrhagic fever [20]. Büyüktuna et al. found no correlation between irisin levels and disease severity or ALT levels [20]. Algül et al. found higher irisin levels in brucella

patients compared to the control group (p<0.05) [18]. In our study, similar to the study conducted by Büyüktuna et al. and Algül et al, we determined that irisin levels (9.27 ng/mL) were higher in patients with brucellosis compared to the control group (4.79, p<0.05). We also observed a positive correlation between irisin levels and increased ALT levels, similar to the study by Rizk et al. these findings of the research indicate a positive association between irisin levels and positive blood culture [19]. This is the first finding in the literature. We confirm that irisin levels are elevated in those patients with brucellosis in our study. In other words, irisin levels increase in infectious diseases. This is due to its anti-inflammatory properties that help limit excessive inflammation [10].

Presepsin regulates cellular and humoral immunity through direct interaction with T and B cells. It increases in response to bacterial infections and decreases after antibiotic treatment. Therefore, it is thought to be an indicator of activation of the immune cell response to an invading pathogen [11]. It is also suggested to be an early biomarker for sepsis. Recent studies have shown that changes in presepsin levels are a suitable indicator for monitoring antibiotic therapy, which improves prognosis and increases survival in cases of severe sepsis or septic shock [21]. Zhao et al. reported that elevated plasma presepsin levels were associated with increased mortality in acute respiratory distress syndrome (ARDS) in the early detection of sepsis-related ARDS [22]. Presepsin has been shown to be a very strong predictor of mortality in intensive care patients with high sensitivity and specificity [23]. In our study, there was no significant difference in presepsin levels compared to the control group. There was a positive correlation between presepsin and irisin levels. This result has been attributed to the slower course of the inflammatory response in brucellosis as compared to other diseases and to limited number of patients in our study.

Apelin, a multifunctional cytokine, has a variety of physiological functions and pathophysiological effects in mammals, including regulating immune responses to improve outcomes in sepsis [24]. ELMeneza et al. investigated serum apelin levels in children with early-onset neonatal sepsis, and they determined apelin levels were significantly higher in patients with early-onset neonatal sepsis with respect to control groups [25]. We did not find a significant difference in serum apelin levels in brucellosis patients compared to the control group. This result was attributed to the limited number of patients in our study.

## Conclusions

Serum irisin levels were higher in brucellosis patients as compared to the population with healthy people. Irisin may play an important role in the inflammatory process and bone marrow suppression in brucellosis patients. Irisin can be used as a diagnostic marker of brucella infection and may be of great benefit in guiding clinicians to predict disease severity and treatment response. However, due to the complexity of the inflammatory response induced by brucellosis, a single biomarker is unlikely to be sufficient in clinical practice. Evaluation in combination with other brucella diagnostic tests may be considered appropriate.

## Author Contributions

**Conceptualization:** Revşa Evin Canpolat Erkan, Recep Tekin.

**Data curation:** Revşa Evin Canpolat Erkan, Recep Tekin.

**Formal analysis:** Revşa Evin Canpolat Erkan, Recep Tekin.

**Funding acquisition:** Recep Tekin.

**Investigation:** Revşa Evin Canpolat Erkan, Recep Tekin.

**Methodology:** Revşa Evin Canpolat Erkan, Recep Tekin.

**Project administration:** Revşa Evin Canpolat Erkan, Recep Tekin.

**Resources:** Revşa Evin Canpolat Erkan, Recep Tekin.

**Software:** Revşa Evin Canpolat Erkan, Recep Tekin.

**Supervision:** Revşa Evin Canpolat Erkan, Recep Tekin.

**Validation:** Revşa Evin Canpolat Erkan, Recep Tekin.

**Visualization:** Revşa Evin Canpolat Erkan, Recep Tekin.

**Writing – original draft:** Revşa Evin Canpolat Erkan, Recep Tekin.

**Writing – review & editing:** Revşa Evin Canpolat Erkan, Recep Tekin.

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
