## [Decision Letter · Decision Letter 0]

17 Nov 2023

PONE-D-23-34264Investigation of New Inflammatory Biomarkers in Brucella PatientsPLOS ONE

Dear Dr. Canpolat Erkan,

Thank you for submitting your manuscript to PLOS ONE. After careful consideration, we feel that it has merit but does not fully meet PLOS ONE’s publication criteria as it currently stands. Therefore, we invite you to submit a revised version of the manuscript that addresses the points raised during the review process. Be sure to:Indicate which changes you require for acceptance versus which changes you recommendAddress any conflicts between the reviews so that it's clear which advice the authors should followProvide specific feedback from your evaluation of the manuscriptPlease ensure that your decision is justified on PLOS ONE’s publication criteria and not, for example, on novelty or perceived impact.

We look forward to receiving your revised manuscript.

Kind regards,

Zhaoqing Du, Ph.D

Academic Editor

PLOS ONE

Journal Requirements:

https://onlinelibrary.wiley.com/doi/10.1111/ijd.12959

https://cyberleninka.org/article/n/985393

https://ijponline.biomedcentral.com/articles/10.1186/s13052-014-0095-1

https://dergipark.org.tr/en/pub/cbusbed/article/887818

In your revision ensure you cite all your sources (including your own works), and quote or rephrase any duplicated text outside the methods section. Further consideration is dependent on these concerns being addressed.

"This publication was supported by Dicle University Scientific Research Projects Commission, dated 11.08.2021, with project number TIP.20.037."

Reviewers' comments:

Reviewer's Responses to Questions

**Comments to the Author**

1. Is the manuscript technically sound, and do the data support the conclusions?

Reviewer #1: Yes

Reviewer #2: Partly

2. Has the statistical analysis been performed appropriately and rigorously? 

Reviewer #1: Yes

Reviewer #2: Yes

3. Have the authors made all data underlying the findings in their manuscript fully available?

Reviewer #1: Yes

Reviewer #2: Yes

4. Is the manuscript presented in an intelligible fashion and written in standard English?

Reviewer #1: Yes

Reviewer #2: Yes

5. Review Comments to the Author

Reviewer #1: Dear author

I think it is a good study that will contribute to the literature. However, my suggestions regarding the study are below:

1. Abstract: It would be appropriate to write the full value of p instead of the p value of iris "<0.05" in the results in the abstract.

2. Materials and methods:What are the diseases or conditions in which the levels of the studied biomarkers are affected? Were these taken into consideration in the patients included in the study?

3. Results: It is recommended that the p value of irisin be written exactly like that of other biomarkers in Table 1. The exact p values of other biomarkers are written, but irisin is written as p < 0.05.

4. Discussion: What could be the reasons why presepsin was not statistically significant in your study? It is recommended that you write your reasons in the discussion section. It seems that some of the studies conducted with presepsin were not significant due to reasons such as the patients' comorbidities not being standardized. Additionally, the number of patients included in the study may also change the significance.

Reviewer #2: 1. Exclusion criteria to be included in the study; Co-infections/ Co-morbidity cases are not mentioned anywhere in the article.

2. Control selection is partially met. Along with healthy controls, other febrile illness groups could have been enrolled. This gives appropriate conclusions for the results.

6. PLOS authors have the option to publish the peer review history of their article (what does this mean?). If published, this will include your full peer review and any attached files.

Reviewer #1: No

Reviewer #2: No

---

## [Author Response · Author response to Decision Letter 0]

12 Dec 2023

Response to reviewer 

We have corrected our article in accordance with reviewer recommendations.

Journal Requirements:

1. Please ensure that your manuscript meets PLOS ONE's style requirements, including those for file naming

Response: Our manuscript meets PLOS ONE's style requirements, including those for file naming.

2. We suggest you thoroughly copyedit your manuscript for language usage, spelling, and grammar. 

Response: We corrected us thoroughly copyedit our manuscript for language usage, spelling, and grammar.

https://onlinelibrary.wiley.com/doi/10.1111/ijd.12959

https://cyberleninka.org/article/n/985393

https://ijponline.biomedcentral.com/articles/10.1186/s13052-014-0095-1

https://dergipark.org.tr/en/pub/cbusbed/article/887818

Response: We are corrected some minor occurrence of overlapping text with the some publication. We are co-authors on some of these publications. The similarities in the articles are due to the same ELISA method being used.

4. Thank you for stating the following financial disclosure: "This publication was supported by Dicle University Scientific Research Projects Commission, dated 11.08.2021, with project number TIP.20.037." Please state what role the funders took in the study. If the funders had no role, please state: ""The funders had no role in study design, data collection and analysis, decision to publish, or preparation of the manuscript."" If this statement is not correct you must amend it as needed. Please include this amended Role of Funder statement in your cover letter; we will change the online submission form on your behalf.

Response: We added this sentence ‘The funders had no role in study design, data collection and analysis, decision to publish, or preparation of the manuscript’

Response: We have included our ethical statement in the Methods section of our article. We deleted ethical statement in the discussion section of our article.

6. Please review your reference list to ensure that it is complete and correct. 

Response: We reviewed our reference list to ensure that it is complete and correct

Reviewer 1:

1. Abstract: It would be appropriate to write the full value of p instead of the p value of iris "<0.05" in the results in the abstract.

Response: We changed p value of the irisin as ‘p=0.045’

2. Materials and methods: What are the diseases or conditions in which the levels of the studied biomarkers are affected? Were these taken into consideration in the patients included in the study?

Response: It is known that different diseases such as gastrointestinal or nutritional disorders, metabolic diseases can significantly affect the levels of irisin, presepsin and apelin. Therefore, persons with any disease including respiratory, endocrinology, cardiac, metabolic or renal diseases and those with viral hepatitis were not included in the study.

We added this sentence ‘Patients with respiratory, endocrine, cardiac, metabolic, renal or hepatic disease and pregnant women were also excluded’ in the Methods section.

3. Results: It is recommended that the p value of irisin be written exactly like that of other biomarkers in Table 1. The exact p values of other biomarkers are written, but irisin is written as p < 0.05.

Response: We changed p value of the irisin as ‘p=0.045’ in the Table 1. 

4. Discussion: What could be the reasons why presepsin was not statistically significant in your study? It is recommended that you write your reasons in the discussion section. It seems that some of the studies conducted with presepsin were not significant due to reasons such as the patients' comorbidities not being standardized. Additionally, the number of patients included in the study may also change the significance.

Response: We had written some sentences in the discussion section for this reason. 

And we added this sentence ‘This result has been attributed to the slower course of the inflammatory response in the brucellosis as compared to other diseases and the limited number of patients in our study’ in the discussion section.

Reviewer 2:

1. Exclusion criteria to be included in the study; Co-infections/ Co-morbidity cases are not mentioned anywhere in the article.

Response: It is known that different diseases such as gastrointestinal or nutritional disorders, metabolic diseases can significantly affect the levels of irisin, presepsin and apelin. Therefore, persons with any disease including respiratory, endocrinology, cardiac, metabolic or renal diseases and those with viral hepatitis were not included in the study.

We added this sentence ‘Patients with respiratory, endocrine, cardiac, metabolic, renal, hepatic disease and pregnant women were also excluded’ in the Methods section.

2. Control selection is partially met. Along with healthy controls, other febrile illness groups could have been enrolled. This gives appropriate conclusions for the results.

Response: This study was designed for patients with Brucella. In the future, we plan to conduct this study with a larger number of patients and more different patient groups.

---

## [Editor Report · Decision Letter 1]

9 Jan 2024

Investigation of New Inflammatory Biomarkers in Patients with Brucella

PONE-D-23-34264R1

Dear Dr. Revşa Evin Canpolat Erkan,

We’re pleased to inform you that your manuscript has been judged scientifically suitable for publication and will be formally accepted for publication once it meets all outstanding technical requirements.

Kind regards,

Zhaoqing Du, Ph.D

Academic Editor

PLOS ONE

---

## [Editor Report · Acceptance letter]

7 Feb 2024

PONE-D-23-34264R1 

PLOS ONE

Dear Dr. Canpolat Erkan, 

I'm pleased to inform you that your manuscript has been deemed suitable for publication in PLOS ONE. Congratulations! Your manuscript is now being handed over to our production team.

Kind regards, 

on behalf of

Dr. Zhaoqing Du 

Academic Editor

PLOS ONE